# 1 Effective earthquake early warning systems: Appropriate messaging
# 2 and public awareness roles

Meng Zhang[1], Xue Qiao[2,3], Barnabas C. Seyler[1], Baofeng Di[1,4], Yuan Wang[1], Ya Tang[1,3]
[1] Department of the Environment, College of Architecture and Environment, Sichuan University, Chengdu 610065, China
[2] Institute of New Energy and Low-carbon Technology & Healthy Food Evaluation Research Center, Sichuan University,
Chengdu, China
[3] State Key Laboratory of Hydraulics and Mountain River Engineering, Sichuan University, Chengdu 610065, China.
[4] Institute for Disaster Management and Reconstruction, Sichuan University, Chengdu 610200, China.
*Correspondence to*: Ya Tang (tangya@scu.edu.cn)
**Abstract.** The earthquake early warning systems (EEWSs) in China have achieved great progress, with warning alerts being
successfully delivered to the public in some regions. We examined the performance of the EEWS in China's Sichuan
Province during the 2019 Changning Earthquake. Although its technical effectiveness was tested with the first alert released
10 s after the earthquake, we found that a big gap existed between the EEWS's message and the public's response. We
highlight the importance of EEWS alert effectiveness and public participation for long-term resiliency, such as delivering
useful alert messages through appropriate communication channels and training people to understand and properly respond.

## 16 1 Why are earthquake early warnings important?

An earthquake is an intense shaking of the Earth's surface, caused by the sudden movement of a plate in the Earth's crust.
Destructive earthquakes, such as the 2008 Wenchuan Earthquake ($M_w$ 7.9) in China, the 2010 Haiti Earthquake ($M_w$ 7.0),
and the 2011 Tohoku-Oki Earthquake ($M_w$ 9.0) in Japan, trigger multiple secondary hazards (e.g., landslides, tsunamis, and
Natech disasters). These earthquakes cause millions of deaths, widespread property damage to buildings and infrastructure,
and severe regional economic fallout. Earthquakes are impossible to avoid, and predicting their occurrence remains difficult,
so more and more countries have focused on developing earthquake early warning (EEW) and emergency management
systems.
An EEW is the detection and characterization of earthquakes as they occur with rapid delivery of alerts to areas potentially
affected before the strongest shaking begins (Allen and Melgar, 2019). Because most of an earthquake's energy is carried by
the damaging S- and surface waves, which arrive after the faster and lower amplitude P-waves, EEW is possible because
both waves travel far slower than the electromagnetic waves used to transfer information (Cremen and Galasso, 2020).
Although the potential warning time may only be seconds to minutes, this time is precious so that individuals and institutions
(e.g., airports, trains, manufacturing, and energy facilities) can take action to save lives and mitigate the potential damage
from earthquakes (Strauss and Allen, 2016).

## 31 2 EEW systems and their applications

Generally, EEW systems (EEWSs) are real-time information systems that consist of three modules, including: 1) monitoring
and detecting earthquakes based on seismic networks; 2) EEW processes, e.g., estimation of location, magnitude, maximum
seismic intensity, and earliest arrival time, as well as alert notification decisions; and 3) information delivery (Cremen and
Galasso, 2020). The importance of EEWSs for disaster mitigation has been widely studied. Many jurisdictions have
operational systems to deliver alerts to the general public (e.g., Mexico, Japan, and South Korea), or target specific
stakeholders in limited areas (e.g., United States, Turkey, Romania, and India) (Allen and Melgar, 2019, and references

therein). There are also some EEWSs in the preparation and testing stages, including in Switzerland, Italy, China Mainland, Nicaragua, and Chile (Allen and Melgar, 2019, and references therein).

Although the theory of EEWSs is simple, the implementation is much more complicated (Allen and Melgar, 2019). An effective EEWS must accurately provide estimated earthquake parameters with long enough warning time to be of practical use for recipients where possible damages may occur. Therefore, most research over the last three decades has focused on evaluating the systems and optimizing their algorithms with the goal of enhancing the quality and accuracy of EEWs. However, several technical challenges are revealed by reviewing the EEWS development (Allen et al., 2009; Allen and Melgar, 2019; Cremen and Galasso, 2020; Hoshiba and Ozaki, 2014; Kamigaichi et al., 2009). For example, 1) it is hard to provide timely warnings in areas closest to epicentres (e.g., the blind zones); 2) when more than two earthquakes occur in close temporal or spatial proximity, the estimation parameters become hard to process and the error substantially increases; 3) the unsaturated magnitude and seismic intensity of large earthquakes (M>8) may be underestimated, such as the Tohoku-Oki Earthquake (Hoshiba and Ozaki, 2014); and 4) the EEWSs may not work properly due to power failures, wiring disconnects, and high background noise caused by large earthquakes and their aftershocks.

Recently, more and more scholars have devoted attention to increasing EEWS effectiveness through social means (e.g., Santos-Reyes, 2019; Sutton et al., 2020), which can alleviate the limitations that are difficult to solve with technical innovations. For example, Japan's EEWS has significantly contributed to reducing social vulnerability to earthquakes through nationwide participation. Most of the alerted respondents could understand and act to protect themselves due to their previous education and training, although the magnitude of the 2011 Tohoku-Oki Earthquake was under-estimated due to technical limitations, resulting in poor-quality alerts (Fujinawa and Noda, 2013; Hoshiba and Ozaki, 2014). In addition, the United States' EEWS (ShakeAlert) enables recipients to immediately participate in the alert process and define the system capability to enhance public participation, which is currently being tested in California, Oregon, and Washington states (Allen and Melgar, 2019). Comparatively, Mexico's EEWS detected and issued warnings for the 2017 Puebla Earthquake; however, the public took a negative attitude towards its performance since they received little information about either the EEWS or the warnings themselves and had not been previously educated how to act during an emergency situation (Santos-Reyes, 2019). These events demonstrate the importance of EEWSs, but also show the critical importance of public awareness education and training before an earthquake occurs, to activate the full benefits of EEWSs.

## 3 China's EEWS Development

China's EEWS development is particularly challenging because multiple regions are prone to earthquakes, including major metropolitan areas. Therefore, following the 2008 Wenchuan Earthquake, China's central government encouraged the establishment of a national EEWS, initially focusing efforts on four seismic regions for pilot testing (**Fig. 1a**). With support from the "National System for Fast Seismic Intensity Reporting and Earthquake Early Warning Program" led by the China Earthquake Administration (CEA), a high-quality national seismological network was installed with 15,000 stations, 1,928 seismic stations (equipped with collocated broadband seismometers and force-balanced accelerometers), 3,114 strong-motion stations (equipped with force-balanced accelerometers), and 10,349 low-cost micro-electro-mechanical system (MEMS)-based sensors (Peng et al., 2020). The instruments aimed at quickly reporting earthquake intensities and earthquake early warnings in key areas on the minute and second scales, respectively. EEWSs in the pilot regions (e.g., the Beijing capital region, southeastern coastal areas, north-south seismic belt, and northern Xinjiang) are now operational and have proven technologically effective to some degree (e.g., physical networks, algorithms, software). Detailed descriptions can be found in Peng et al. (2011), Peng et al. (2020), and Zhang et al. (2016), but few of these studies have focused on the information dissemination mechanisms and public perception to the EEWS.

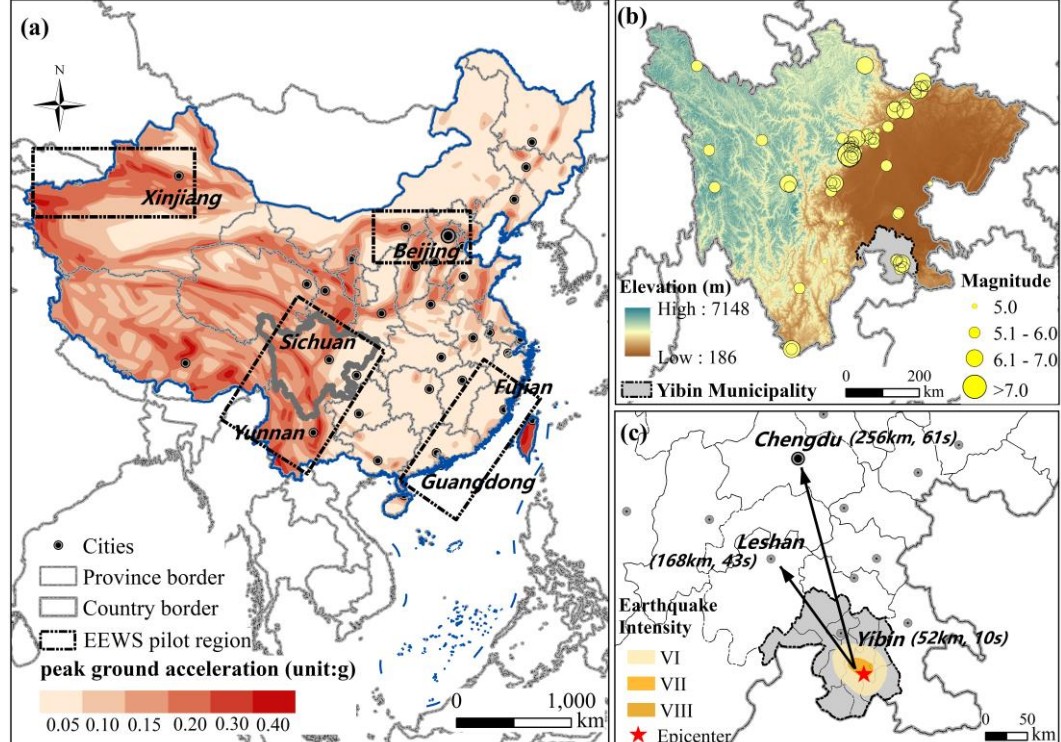

78

**Figure 1** Seismic activity and EEWSs across China. (a) Distribution of earthquake intensity and EEWSs in various Chinese regions (modified from seismic peak ground acceleration zonation map of China; Standardization Administration of the People's Republic of China, 2015); (b) historical earthquakes (January 1949-August 2020) in Sichuan Province; and (c) location of the Changning Earthquake.

## 3.1 Fujian case

As one of the main pilot areas, a provincial EEWS was first built in Fujian in 2009, with 125 seismic monitoring stations (equipped with velocity and acceleration meters) across the whole region with an average distance of 31 km between them. Each station connects to the Fujian Earthquake Agency (FJEA) for EEW processing and information release through dedicated optical fiber cables provided by China Telecom. The preparatory process of Fujian's EEWS included two steps: 1) design the EEWS and test its technical capabilities (Zhang et al., 2016); and 2) design the content and criteria for issuing EEW alerts to the public (Zhang et al., 2016), which is similar to the development of Japan's EEWS (Kamigaichi et al., 2009). The FJEA can issue alerts and authorized third parties can forward these alerts through multiple channels, including broadcast (television and radio), special terminals, Internet, and smartphone Apps[1]. EEW alert receiving terminals are preferably installed in schools, factories, and residential communities, especially those in high earthquake risk areas, where the coverage rate must be greater than 60%. More importantly, when, what, and how to deliver an EEW alert is regulated by provincial standards[2]. For example, only when the predicted seismic intensity is greater than VI (Chinese intensity scale)[3], will FJEA warn the provincial public with red or orange signal icons (I and II EEW) and sounds. Fujian's EEWS began issuing alerts to the public in 2017, and Fujian's successful model was later extended to other regions in China.

## 3.2 Sichuan case

Sichuan is a major earthquake-prone region. Based on the China Earthquake Networks Center (CENC, http://www.ceic.ac.cn/history), 73 earthquakes above magnitude Ms 5.0 occurred in Sichuan between the 2008 Wenchuan Earthquake and April 1, 2020 (**Fig. 1b**). A hybrid demonstration EEWS was built in the border region between Sichuan and

---

[1] The mobile app Earthquake Warning in Fujian can be downloaded at http://www.fjdzj.gov.cn/ar/2018050814000013.htm.

[2] Release of earthquake warning information (DB35/T 1666-2017). (In Chinese)

[3] The Chinese seismic intensity scale (GB/T 17742-2008).
http://c.gb688.cn/bzgk/gb/showGb?type=online&hcno=AE2DAA79A7404FFAC73A9F3A33FBAA5A (In Chinese)

Yunnan provinces in 2015, with 270 MEMS-based stations, as a part of China's EEWS (Peng et al., 2020). The real-time
data recorded by these stations can be transferred through 3G/4G mobile network to the Sichuan Earthquake Administration
(Peng et al., 2020). In contrast to the hybrid demonstration EEWS introduced in Peng et al. (2020), Sichuan's EEWS is
operated by a third-party (Institute of Care-Life, ICL) in collaboration (at the municipality and county level) with the
Emergency Management Bureau (Wang and Lin, 2020). The recent $M$s 6.0 Changning Earthquake happened at 22:55 PM on
17th June 2019 in southeast Sichuan's Yibin Municipality, triggering an alert in some cities across the province, including
Yibin (52 km from epicenter), Leshan (168 km), and Chengdu (245 km) (**Fig. 1c**). The alerts were issued approximately 10 s,
43 s, and 61 s prior to major shaking in the above cities, respectively. It was the first time that an EEWS alert was triggered
to the general public in Sichuan, which generated great public interest and confusion.
In Chengdu, the provincial capital city, the alert was delivered in several ways, including broadcast sirens, as well as text
messages on televisions and cell phones that had special applications installed. The broadcast siren notified the most people
with speakers located in more than 110 residential areas. The alert began with a countdown, followed by loud alarm sirens.
However, few people understood what the siren pertained to or what was about to happen with only a countdown and then
siren. Only when the shaking began did most people realize the alarm was intended to warn of an impending earthquake.
Most people reported that when the countdown began over broadcast speakers followed by the siren, they were confused and
unsure what to do. They did not know what was happening or what would happen, because the countdown and siren were
unaccompanied by clear audio messages with explanatory information. Many people interpreted the alarm as a firemen's
duty task, an air raid alert test, an explosion, theft alarm from a car or electric bicycle, or a special sales event. Clearly, due to
the diversity of reactions, the alert caused more confusion, fear, and disturbance than what was intended by the EEWS. Some
people were less concerned with the earthquake than by the confusion over the loud countdown and siren, as it was nearly
midnight.
We examined the public perception of Sichuan's EEWS during the Changning Earthquake through an internet-based survey
conducted June 21-23, 2019, in Chengdu. The online questionnaire was administered by the survey platform Wenjuanxing
(https://wjx.cn) and delivered to the public via social media (WeChat). We received a total of 770 responses. The survey
contained 11 questions in total, with 9 quantitative (single choice) and 2 qualitative (free response) questions, related to
demographics, earthquake preparedness and knowledge, behavioural responses to EEW alerts, and reasons for those
responses (see questionnaire in the Supplement). Survey respondents were asked whether or not they had heard the sirens on
the day of the earthquake, and based on their response, the participants were divided into two groups: 1) those who heard the
broadcast siren alert in real time (Group A, n=261) and 2) those who did not (Group B, n=509). Although participants in
Group B had not heard the sirens on the day of the earthquake, both groups were shown a video of the siren/alert at the time
of the survey to detect their behavioural responses to the sirens. The descriptive information, basic frequency, and cross-
tabulation analyses of the collected data were undertaken using SPSS software (Lee Abbot and Mckinney, 2013). For cross-
tabulations, statistical significance was determined using the Pearson Chi-Square test.
Demographics of respondents can be found in **Table 1**. We separately tested for differences between the two independent
sample populations for each response. The results (**Fig. 2**) show that large pluralities of both groups (Group A, 41%; Group
B, 45%; $p < 0.001$) did not understand the purpose of the alert and felt confused or scared by it. The proportion of
respondents from both groups who stated that they understood the alert but did not know how to react was the same (7% vs.
7%, $p<0.001$). Surprisingly, a significantly larger proportion of respondents from Group B understood and knew what
actions to take (32%, $p<0.001$) than Group A (21%, $p<0.001$). Of those from Group A who knew what actions to take, their
knowledge came primarily from previous training (26), hearing a brief note at the beginning of the alert (11), being informed
by people nearby when the alert was ongoing (7), or for several other reasons (11). Because so few people knew what the

alert was about or recognized what was about to happen, most people did not have sufficient knowledge or awareness of the correct actions to take. Consequently, this alert could have caused additional problems, including injuries or cardiovascular problems due to fear or panic from the sudden high-decibel sirens blaring over loudspeakers, and the resulting confusion could also have led to more acute harm if the shaking level had been higher.

We also tested the role demographic variables (e.g., gender, age, and occupation) may have on predicting how the public may respond to EEWs and their earthquake awareness. The results (Table S1) show that both gender and occupation were significantly associated with how the public responds to earthquake warnings. From our sample results, it appears males and people holding certain occupations (e.g., governmental organizations and emergency institutions) were more likely to have already received the type of pre-earthquake training necessary for them to know how to respond to the earthquake warnings. If this is true, special effort should be made to target those segments of the population that were under-prepared. However, due to the likelihood of self-selection bias in our sample, more research is necessary to verify and further explore the implications of these findings so as to better inform policy and guide future pre-earthquake preparedness efforts.

**Table 1** Demographic profile of internet-based survey participants regarding responses to early warning of Changning earthquake (N=770)

| Variable | | N | % |
|---|---|---|---|
| Gender | Male | 220 | 28.6 |
| | Female | 550 | 71.4 |
| Age | ≤18 | 5 | 0.6 |
| | 19-30 | 204 | 26.5 |
| | 31-40 | 326 | 42.3 |
| | 41-60 | 165 | 21.4 |
| | >60 | 70 | 9.1 |
| Education level | Primary or below | 28 | 3.6 |
| | High school | 69 | 9.0 |
| | Undergraduate | 491 | 63.8 |
| | Postgraduate | 182 | 23.6 |
| Occupation | Students, educational employees, and academics | 167 | 21.7 |
| | Governmental organizations | 58 | 7.5 |
| | Emergency institutions and companies | 97 | 12.6 |
| | Private business and farmers | 330 | 42.9 |
| | Other | 118 | 15.3 |
| Earthquake training and education | Yes | 518 | 67.3 |
| | No | 252 | 32.7 |

Note: The category of emergency institutions and companies refer to those that typically require earthquake alerts, such as hospitals, railways, and factories with hazardous environments. The category of "others" included those without formal jobs and retirees.

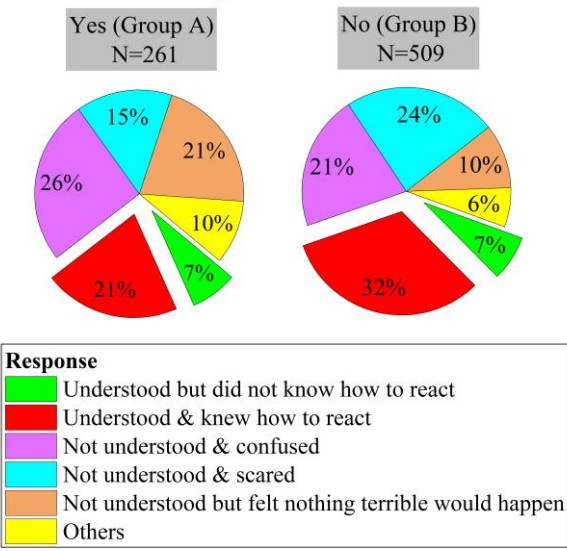

**Figure 2** Public responses to the siren/broadcast speaker of early warning for the Changning Earthquake from an internet-based survey in Chengdu, China.

## 4 EEWS Limitations and Implications from Sichuan

The Changning Earthquake's example highlights some challenges with Sichuan's EEWS. We are not arguing against issuing earthquake alerts, however, this event and the resulting confusion raises four important issues that should be addressed moving forward.

First, a big gap exists between the intention of an EEWS and its reality in Sichuan. The most important intended effect of an EEW is to enable residents to take protective actions within the short time before the shaking arrives (Nakayachi et al., 2019). Only when an EEWS is sufficiently tested and widely publicized (Kamigaichi et al., 2009) can people understand the meaning of an alert and take appropriate actions (Kamigaichi et al., 2009). When installed in a residential area, inhabitants should be notified about the system, and most importantly, informed about what actions they should take after receiving an alert, but before shaking begins. In the case of the EEWS's alert in Chengdu following the Changning Earthquake, inadequate efforts had been made to inform the public prior to the earthquake, so few people were able to understand or respond appropriately to the alert. The experience of countries like Japan shows that public training, education, and widespread awareness campaigns about EEWSs are the key factors to their success (e.g., Fujinawa and Noda, 2013; Kamigaichi et al., 2009).

Second, of vital importance is what and how to deliver actionable warnings to the public. An effective early alert should not only inform the public about hazards, but also protective actions (Allen and Melgar, 2019; Sutton et al., 2020). The default messages must be simple, because the content and comprehension of EEW messages should result in people taking appropriate actions (Allen and Melgar, 2019; Becker et al., 2020a; Santos-Reyes, 2019). Messages can be instructions (e.g., Drop, cover, and hold on; US), origin time, and names of epicenter regions and subprefecture areas (e.g., Earthquake early warning. An earthquake has occurred in Area X. Please prepare for strong temblor; Japan) (Kamigaichi et al., 2009; Allen and Melgar, 2019). Providing information about expected shaking intensity or arrival time (countdown) are not recommended, as these can lead to unnecessary panic (Allen and Melgar, 2019; Kamigaichi et al., 2009), but some studies hold the opposite viewpoint (Santos-Reyes, 2019). Furthermore, the information and alerts should be delivered in stable, useful, and suitable ways. As our case study shows, some claimed that the earthquake itself did not scare them as much as the blaring siren did. It seemed unnecessary to use sirens on loudspeakers that day, especially during the night. While the

advantage of using sirens is that it rapidly reaches people simultaneously, the use of such "shocking" alarms is needed only with high risks and likelihood of considerable damage. For those that may not lead to causalities or considerable social or economic losses, use of more "gentle" alert channels is recommended. Alerts delivered over the radio, TV, SMS messages, emails, and smartphone applications have shown greater effectiveness in documented cases (Hoshiba and Ozaki, 2014).

Third, at what level of seismic intensity the alert should be triggered is a key issue. It is essential to avoid the fabled "boy crying wolf" or over-alerting, which can lead to public frustration and apathy, so alert messages should not be issued unless the shaking is expected to cause considerable damage. The Changning Earthquake did not cause strong motion or significant damage in Chengdu, but 15% and 24% of the participants from Groups A and B (**Fig. 2**) were terrified by the alarm sound, respectively. At the time, Sichuan did not have specific criteria for when to issue EEW alarms. The provincial standard was only issued in April 2019, so it had not yet been formally implemented. According to this standard (draft version)[4], a warning should only be issued (to the general public) when the seismic intensity is expected to be VI on the Chinese scale. However, despite the higher level in Yibin, the seismic intensity in Chengdu was lower than VI (**Fig. 1c**), so the alert should not have been issued in Chengdu. In addition, there continues to be insufficient guidance about how to handle false alarms, updates, and canceled warnings.

Fourth, earthquake alerts should be released by an authoritative government agency. The public should be informed that only alerts from the authorized body are reliable. But it was unclear who released the alert on June 17, 2019. There can be many third-party warning service providers, who forward EEW messages by multiple transmission routes. Yet, according to Sichuan's draft standard, the publishing body should only be the Provincial Earthquake Warning Release Center. In addition, the Sichuan case shows that one region may have multiple EEWSs (Wang and Lin, 2020), which will raise greater challenges regarding best practices for issuing EEW and popularizing how to interpret them. Therefore, greater supervision and management systems are urgently needed in Sichuan's EEW practice.

The most important component of a successful EEWS is a group of users with awareness and preparedness, who want alerts and will take protective actions (Allen and Melgar, 2019). The next is the physical infrastructure and sensor system (Allen and Melgar, 2019). The Changning Earthquake warning event showed that the transmission and utilization of the EEW lagged behind the technological development and physical construction. Moreover, the public in affected areas were not well-informed by EEWS alerts, nor were they adequately trained on how to respond. Therefore, we highlight the successful public education and preparedness training model from Japan's seismic culture, because the relatively poor understanding of an EEWS by the public can result in confusion. Useful strategies include: 1) launching education programs on what actions should be taken before, during (at various timeframes), and after an earthquake (Santos-Reyes, 2019). Research indicates that alert messages with guidance on actions may be useful as a reminder to achieve optimal behavioral responses, but only when people are already familiar with these actions prior to receiving a warning (Becker et al., 2020b); 2) carrying out drills and exercises to improve personal practical skills and earthquake preparedness (Nakayachi et al., 2019), which is particularly important for regions new to EEWSs. Yet, beyond what actions are necessary to take in response to warnings (Ji et al., 2019; Sutton et al., 2020), the public also needs education regarding the technical limitations and accuracy of EEWSs (Kamigaichi et al., 2009). We also suggest that Chinese scholars should focus more effort on the public response to and perception of EEWSs to get more insights for issuing alerts, managing emergencies, and making policy.

---

[4] Sichuan Seismological Bureau organized institutions to complete the drafting of "emergency earthquake information release earthquake warning information". The local standard draft was published for public comments.
http://www.scdzj.gov.cn/jlhd/yjzj/202004/t20200429_54006.html (Accessed on April 29, 2020)

Furthermore, due to differences in geological settings, socio-economic development statuses, and population densities, losses
caused by earthquakes of the same magnitude can vary greatly. Therefore, it is also very important to decide where an
EEWS should be set up. Since earthquakes are disasters faced by many countries, collaboration in development and
application of EEWSs among countries or regions should be encouraged, so that appropriate efforts are made to reduce loss
of life and property when earthquakes occur, despite their inability to reduce losses in epicenter areas.
Several limitations of the present study and current scholarship are as follows. First, our research revealed that what and how
to deliver actionable warnings to the public is of vital importance, but we also found that differences exist even between
countries with relatively mature EEWSs. For example, some research on the public's perception of Mexico's EEWS
highlighted the need to issue warning times (Santos-Reyes, 2019), while other studies thought it was unnecessary (Allen and
Melgar, 2019; Kamigaichi et al., 2009). More work is needed about what and how best to deliver warnings in the Chinese
context. Second, many studies concluded that education and training are crucial for enhancing earthquake preparedness
(Nakayachi et al., 2019; Santos-Reyes, 2019; Sutton et al., 2020), but few tested whether these strategies were useful or not.
Local people's knowledge about earthquake risks as well as their previous training/education about how best to respond
were ascertained in our survey by asking whether respondents had previously received training and/or obtained education.
Nevertheless, the lack of reliable data about dissemination of earthquake awareness training and education materials is a
challenge for these types of studies. Further research efforts should investigate strategies to increase public attention to this
aspect of EEWSs.

## 5 Conclusion

The Changning Earthquake warning event demonstrated that EEWSs are not simply technological engineering infrastructure,
but they are also social systems for disaster mitigation. There will be no substantive benefit without proper knowledge and
appropriate emergency responses by the public, even if the warning is issued accurately and timely, as evidenced by the
experiences of Mexico and Chengdu, China. Although authoritative government agencies have emphasized that information
release services are the "last kilometer" for earthquake warning systems to reach the public, the actual implementation
showed that the "last kilometer" was not obstacle-free. It is worth considering how to best release and effectively convey
early warning information based on China's actual reality, not an idealized situation. The construction of EEWSs, issuance
of alarms to the public, and formation of public awareness by science education are inseparably related. We recommend that
China should collect best practices of EEWS utilization domestically and internationally in cases of EEW alert delivery to
the public for the purpose of more effective promotion of EEWs. Finally, greater collaboration among countries would
benefit many more people around the world.

**Data availability**

The questionnaire data used to support the findings of this study are available upon reasonable request.

**Author contributions**

YT and XQ designed the research. BFD and WY conducted the survey. MZ and XQ performed the data curation, formal
analysis, and wrote the original paper. YT, XQ, and BCS were responsible for supervision. All authors participated in
improving the paper by editing.
**Competing interests**
The authors declare that they have no conflict of interest.
**Acknowledgement**
This study was supported by Department of Science and Technology of Sichuan Province (2020YFH0023) and Specialized
Fund for the Post-Disaster Reconstruction and Heritage Protection in Sichuan Province (No. 5132202019000128). We
appreciate the contribution of the China Earthquake Administration (http://data.earthquake.cn/), Fujian Earthquake Agency
(http://www.fjdzj.gov.cn/), and Sichuan Earthquake Administration (http://www.scdzj.gov.cn/).

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
