# Peer review of "Effective earthquake early warning systems: Appropriate messaging"

_Natural Hazards and Earth System Sciences, 2021_

## Author Response (AR1)

July 6, 2021

Daniela Molinari, PhD
*Natural Hazards and Earth System Sciences*

Dear Dr. Molinari,

Thank you for the opportunity to revise our manuscript (nhess-2021-42), newly entitled "Effective earthquake early warning systems: Appropriate messaging and public awareness roles". We appreciate the helpful and insightful comments from the editor and peer reviewers.

As you have noted, we have carefully considered and responded to each reviewer's comments point-by-point during the peer review process, which can also be found as an annex to this letter. In this major revision of our manuscript, we have added additional details about the monitoring, detection, and delivery of warning messages (Lines 67-72, 83-95, 99-101), as well as the differences between the responses of the two groups in our survey (Lines 132-137; Figure 2). Although we briefly introduced the contents of warning messages (e.g., Fujian case, Lines 93-95; Sichuan case, Lines 110-111), our study showed and discussed that appropriate messaging is critical for effective earthquake early warning systems. As to the comments related to the training/public information (e.g., survey questions in the Supplement) and representativeness of the sample, we had explained and responded during the interactive discussion that the lack of reliable data about dissemination of earthquake awareness materials is a challenge for these types of studies. As per your request, we have prepared both revised and marked-up versions of our manuscript, which we are uploading to the reviewers as the authors reply.

The goal of our study was to assess the relative effectiveness of the messaging in terms of public understanding/perception of the EEWS in Chengdu. We found a big gap existed between the EEWS's messaging and the public's response following the Changning Earthquake. We, therefore, chose to submit this study as a "brief communication" due to the timeliness of the topic and the significance of advancing understanding of a relatively focused but important research question in the context of China (but with implications for other countries as well). We appreciate your constructive feedback, but respectfully disagree with your suggestion of resubmitting our paper as a research article, because this brief communication has already been reviewed freely for a long time on the internet, it makes no sense at this point to reconfigure our study now as a research paper. As you may realize, our research group has particular goals related to why we submitted this study as a brief communication to your journal. If at this point, we would need to reconfigure our study to meet the requirements of a research paper and then begin the review process from the beginning again, we may decide instead to submit our

manuscript to another journal. We hope you understand we chose your journal and this manuscript type with particular reasons in mind.

We agree that the comparative analysis of hardware and systems between China/Sichuan and other countries is interesting and thought-provoking. As we explained previously in our response to the reviewers, previous studies have already introduced, summarized, and compared the design and utility of EEWSs from the technical perspective (Allen and Melagr, 2019; Cremen and Galasso, 2020; Peng et al., 2011, 2020; Zhang et al., 2016). Our study did not compare the difference of EEWSs in China/Sichuan and other countries due to: (1) our goals to rapidly get this message out in the form of a brief communication, which necessarily constrains the length of a manuscript; (2) more importantly, we believe that the comparative discussion in our paper on how to better deliver alerts and increase awareness is more meaningful, rather than duplicating what other influential studies have already done.

We believe these revisions have significantly improved this manuscript and look forward to this opportunity to be considered for publication in *Natural Hazards and Earth System Sciences* as a "brief communication".

Please feel free to contact me with any questions or concerns you may have.

Sincerely,
* * *
Ya Tang, PhD

Department of Environment
College of Architecture and Environment
Sichuan University, Chengdu, China
tangya@scu.edu.cn

**Annex**

**Point-by-point responses to comments from Reviewers**

**Reviewer #1:** This is an important, interesting topic. The paper is clearly written and easy to follow. There are a number of short-comings which should be addressed in a revised version:

We thank you for your helpful suggestions and constructive feedback to help us improve our manuscript. We have responded to your specific comments below (line numbers refer to the clean version).

1. More details on the monitoring and detection; information processing and delivery of information should be better clarified; it is not clear what training and public information was delivered prior to and how often prior to the earthquake event;

**Response:** In the previous version of our manuscript, we introduced the modules in the EEWS (retained in Lines 32-35). We have added additional details on the monitoring, detection, and delivery of China's EEWSs and its pilot regions (e.g., Fujian and Sichuan Province) in our revised manuscript (Lines 69-72, 83-95, 99-101). For more information about technical capabilities, we also recommend the published articles (e.g., Peng et al., 2011, 2020; Zhang et al., 2016) (retained in Lines 75-77).

As to the public training and education, in China, these are usually carried out by the relevant emergency management agencies, earthquake administration bureaus, schools, or research institutions. For example, the webpage of the Sichuan Earthquake Administration (http://www.scdzj.gov.cn/) includes articles, posters, and videos about earthquake preparedness (e.g., earthquake hazards, past events, proper response actions, etc.), but it is difficult to obtain more detailed information (e.g., how often, to whom, and where these materials are disseminated).

For our study, local people's knowledge about earthquake risks as well as their previous training/education about how best to respond was ascertained in our survey by asking whether respondents had received training and/or actively/passively obtained education (Questions 5-6, Supplement). To further clarify this, we have added the survey questions as an appendix to this article. We found that of the 770 respondents to our survey, 518 (67.3%) had received general education about earthquakes and their risks, but only 26 (3.4%) had received specific education related to the EEWS. Nevertheless, the lack of reliable data about dissemination of earthquake awareness and education materials is a challenge for these types of studies, as noted elsewhere (Becker et al, 2020; Santos-Reyes, 2019).

2. More details regarding the survey respondent recruitment, sampling, and representation should be included; is this valid sample?

**Response:** As stated in Paragraph 2 on Page 5, the online questionnaire was administered by the survey platform Wenjuanxing and delivered to the public via social media (WeChat). Therefore, the survey respondents were not randomly selected, so there was likely a self-selection bias in the types of people who responded. Although this does limit the generalizability of our findings to some extent, due to the lack of similar surveys in the immediate aftermath of earthquake events in China, we believe this sample was sufficient to ascertain the general lack of effectiveness of Sichuan's EEWS messaging at the time of this particular earthquake.

3. The statistical analysis is very thin; need to include at a minimum standard test of significance, prob values, confidence interval for the comparisons between groups and categories.

**Response:** We have added the statistical analysis as you suggested. The results with corresponding descriptions can be found in Lines 132-137.

4. Missed opportunities for more analysis of both differences between the characteristics of those who understood and those who did not the EEWS; what is the role of education, social status, age, gender, previous experience, etc.

**Response:** We have compared the differences between the responses of those who understood and those who did not understand the EEWS (Lines 132-137; Figure 2). We agree with the importance of understanding the roles of these variables you mentioned in predicting the level of awareness of the EEWS in Sichuan, however, the survey respondents were not randomly selected, so the likelihood of a self-selection bias in the types of people who responded limits the types of analyses otherwise possible. Therefore, this type of analysis was beyond the scope of this study, but certainly merits further research.

5. More discussion of the implications for improving earthquake warning and how to increase awareness, knowledge, comprehension and actions resulting from EEWS;

**Response:** We appreciate the value of the points you raised, but are unsure of the extent of discussion you see is lacking, since our study found a big gap existed between the EEWS's message and the public's response following the Changning Earthquake. Therefore, our results highlighted the lack of effectiveness of the EEWS messaging prior to the earthquake event. We discussed this at length in our manuscript (Section 4). As a brief communication format article, it is not possible to explore every aspect of these challenges within the word limitation and scope of this paper. Moreover, there is limited research in and outside of China on public perceptions and responses to EEWS messaging, and the published work is limited in the exploration of how to improve their effectiveness (Nakayachi et al., 2019), so we believe our discussion significantly advances current understanding in the field, especially within the context of Sichuan.

6. Comparative analysis between hardware and systems between China/Sichuan and Japan/US/Mexico would be useful;

**Response:** As you know, the designs of EEWSs (e.g., sensors, communications and telemetry, processing capabilities, and transmitters/receivers to deliver alerts) in various countries are quite different depending on each country's very unique context. Previous studies have summarized the utility of EEWSs from the technical perspective, including best approaches, instruments, and algorithms for EEWSs implemented in various countries, providing comprehensive and synthetic insights (Allen and Melagr, 2019; Cremen and Galasso, 2020). As stated in Lines 75-77, the utility of EEWSs (mainly physical networks and algorithms) in China's main pilot regions (e.g., Beijing capital region, Yunnan-Sichuan boarder region, and Fujian) can be found published elsewhere already (e.g., Peng et al. 2011, 2020; Zhang et al. 2016). These studies highlight the importance of assessing technological capabilities of EEWSs in terms of accuracy and latency of alert delivery, so this study did not seek to replicate their findings. However, unlike those studies, the goal of this study was not to provide insights into the hardware or software systems of the EEWSs themselves, but instead to assess the relative effectiveness of the messaging in terms of public understanding/perception of Sichuan's EEWS in order to provide insights to local authorities on how to improve before future earthquake events.

Specifically, given the limited research on the public's perceptions and response to EEWS alerts, we sought to examine the effectiveness of the messaging/alerts for personal protection rather than technological capabilities. Our study sought to examine whether the public could reasonably be expected to take appropriate protective actions with the types of alert messages issued during the 2019 Changning Earthquake. The findings of our survey indicate that the public in Sichuan were not adequately prepared to take appropriate actions based on the types of messaging they had received before and during the earthquake event. Thus, as a brief communication, rather than duplicating what other studies have already done (in terms of comparing the technical hardware/software, etc.), we tried to instead provide more detail about how to better deliver alerts and increase awareness, knowledge, and appropriate actions by comparing the case of Sichuan's EEWS to other regions.

While the paper has potential, in its present form even as a brief communication, it simply raises more questions than it answers...

We appreciate your constructive feedback, but disagree with your conclusion. In a brief communication format, we believe it best to advance understanding of a relatively limited research question (e.g., how effective is Sichuan's EEWS messaging in terms of public understanding), particularly in the context of China, where these types of studies are more limited than in other earthquake-prone regions of the world. Thus, we do not agree that a brief communication-type manuscript should address every interesting and thought-provoking angle

about EEWSs, especially not those that have already been addressed in great detail in previous manuscripts.

**Reviewer #2:** The paper clearly expresses the concepts described as objectives, but there are some parts to be improved/modified.

We are encouraged that you agree that our manuscript clearly expresses the concepts described as our objectives in this study. Based on your comments, we revised our manuscript, and believe this revised version is an improvement. We thank you for your helpful suggestions and constructive feedback.

1. In the title it could be better to add a reference related to the importance of the population preparedness about the EEWS, the second pillar of the paper together with the messages' characteristics. It could be something like: "Effective earthquake early warning systems: appropriate messaging and population preparedness roles".

**Response:** Based on your suggestion, we have revised the title of this manuscript. However, in our study, we found that the main limitation of Sichuan's EEWS was that the public's lack of awareness of the EEWS prevented their understanding of the time-sensitive alert messaging. Since we did not specifically address the broader topic of "public preparedness" in this article but instead the more limited "public awareness" and "education", the new title is: "Effective earthquake early warning systems: Appropriate messaging and public awareness roles".

2. The description of the responders' samples has to be better organized at the beginning of the related paragraph (starting from line 106). As example, it's important to move in this paragraph the sentence written in the note 1, page 7 about the group B, to permit a better understanding of the survey.

**Response:** Based on your suggestion, we have significantly revised and reorganized the paragraph of the respondent samples (Lines 120-131). We added more detailed information about the design and delivery of the Internet-based survey. We reorganized the sentences describing both groups to provide a better understanding of the respondents.

3. Figure 1: add the unit of measure in the legend of fig. 1(a). The caption of the figure is too long. The four regions can be described in paragraph 3, as partially done in line73-75.

**Response:** We have added the unit of measurement in the legend of fig. 1(a) and shortened the caption of the figure. We have moved the description of the four regions to Lines 73-74.

4. There is a paragraph 3.2 but not a paragraph 3.1

**Response:** Thanks for pointing out this oversight on our part. We have corrected the number of the paragraph.

5. I agree with the other comments written by the Referee #1 (09 Mar 2021)

**Response:** We have carefully responded to the specific comments from Referee #1.

6. My comments are strictly related to this paper, and not about the positioning of the paper in the literature about the topic "Earthquake Early Warning Systems".

**Response:** We appreciate your helpful comments.

7. A general revision of the language is suggested.

**Response:** We have carefully proofread the revised manuscript.

---

## Referee Report (RR1)

Journal: NHESS
Title: **Effective earthquake early warning systems: Appropriate messaging and public awareness roles**
Author(s): Zhang et al.
MS No.: NHESS-2021-41
MS Type: Brief communication
**Iteration: Second review**

The paper aims at stressing and discussing the importance of social aspects for EEWS's effectiveness, by critically analysing the ability of people of understanding the received warning, and acting properly, after the earthquake in Chengdu, China, in 2019. After briefly introducing technical aspects of the EEWS implemented in the Sichuan province, the authors discuss the results of a survey aimed at understanding the level of awareness and preparedness of Chengdu citizens, with respect to warning messages. Results reveal important limitations of EEWSs in China, mostly related to people awareness and preparedness; accordingly, authors suggest actions to improve the present situation, by referring to state of art examples.

I have read the revised version of the manuscript and I think that the paper has been improved with respect to the previous version submitted to the journal, with authors addressing most of the issues raised by the previous referees, on which I mostly agree. In fact, I agree with the authors that the main scope of the paper is social rather than technical aspects. For this reason, I support their choice of not investigating and comparing hardware and software systems of cited EEWSs.

I recommend the publication of the paper in this present form.

---

## Author Response (AR2)

September 16, 2021

Daniela Molinari, PhD
*Natural Hazards and Earth System Sciences*

Dear Dr. Molinari,

We would like to thank you for the opportunity to revise and resubmit our manuscript (nhess-2021-42), entitled "Effective earthquake early warning systems: Appropriate messaging and public awareness roles". We appreciate the helpful and insightful comments from the editor and peer reviewers.

In this minor revision of our manuscript, we have carefully considered and responded to each suggestion, which can also be found as an annex to this letter. Based on your suggestion, we have tested the role of demographics variables on public responses to and their knowledge about earthquakes. The results indicate that males and people with certain occupations (e.g., governmental organizations and emergency institutions) may be more likely to have already received the type of pre-earthquake training necessary for them to know how to respond to earthquake warnings. However, as we explain in the revised manuscript, due to the likelihood of self-selection bias in our sample, more research is necessary to verify and explore the implications of these findings. We have made a few changes to some sentences to add greater clarity and also carefully proofread the revised manuscript. As per your request, we have prepared both revised and marked-up versions of our manuscript.

We believe these revisions have significantly improved this manuscript and look forward to this opportunity to be considered for publication in *Natural Hazards and Earth System Sciences* as a "brief communication".

Please feel free to contact me with any questions or concerns you may have.

Sincerely,
* * *
Ya Tang, PhD

Department of Environment
College of Architecture and Environment
Sichuan University, Chengdu, China
tangya@scu.edu.cn

**Annex**

**Point-by-point responses to comments from the editor**

Dear Meng Zhang and co-authors,

Thank you again for the submission of your paper "Brief communication: Appropriate messaging is critical for effective earthquake early warning systems" to NHESS.

The revised version you supplied has been sent to two anonymous referees that agreed on the suitability of your paper for publication, after some technical amendments. Still, as editor, I think that the description of the sample and the statistical analysis you did must be further improved to reach the quality level of the journal, even if it is a brief communication paper. For this reason, I recommended "minor revisions".

We thank you for your helpful suggestions, constructive feedback, and the opportunity to improve our manuscript for publication. We have responded to your specific comments below.

**Section #1:** In detail, I agree with one of the first referee that more details regarding the survey respondent recruitment and representation should be included, stressing that results can be affected by a possible bias due to self-selection (as you explained in your response). I do not agree with you when you state that such a bias limits the analysis of the role of education, social status, gender, etc., so I encourage you to include such kind of analysis in the paper.

In response to the first referee you mentioned, we previously added additional details regarding the survey respondent recruitment and sampling (Lines 122-124). Based on your suggestion, we have tested the role of demographic variables on how the public responds to and their knowledge about earthquakes. The results (Table S1) show that gender and occupation were significantly associated with the public response to earthquake warnings. This may indicate that males and those with certain occupations (e.g., governmental organizations and emergency institutions) are more likely to have already received the type of pre-earthquake training necessary to know how to respond to earthquake warnings. However, due to the likelihood of self-selection bias in our sample, more research is necessary to explore the implications of these findings. We explained these findings and discussed their implications in a new paragraph in our revised manuscript (Lines 145-152) and included the results in a Supplementary Table (S1).

Finally, I would suggest to change Figure 1 with an improved-quality and more significant one.

**Response:** Based on your suggestion, we have modified Figure 1 with improved-quality and higher resolution.

**Section #2:** I also recommend you to provide the technical amendments suggested by the second referee (which I also report below for the sake of simplicity):

1. Line 94: change "can FJEA" with "FJEA can"

**Response:** We have changed "can FJEA" to "will FJEA" to clarify our intended meaning.

2. Line 101-102: the sentence "In contrast to the demonstration EEWS," is not clear

**Response:** We have revised this sentence as "In contrast to the hybrid demonstration EEWS introduced in Peng et al. (2020)" to add greater clarity.

3. Line 109: delete the words "Of these"

**Response:** Done.

4. Line 130: add some reference for SPSS, and add the word "software"

**Response:** Based on your suggestion, we have added a reference for SPSS and also the word "software".

5. Line 133: after "The results" add "(Fig. 2)"

**Response:** Done.

6. Line 153: "alerts. However" change point with comma

**Response:** Done.

7. Line 154: change colon with point at the end of the sentence, or use bullet points for the four issues

**Response:** Done.

8. Line 157: change "can people" with "people can"

**Response:** This sentence grammatically is an "Only when A can B happen" construct (e.g., Only when an EEWS is sufficiently tested […] can people understand the meaning of an alert […]). One of our co-authors is a native English speaker and disagrees, respectfully, with this suggestion.

9. Line 180: add "of seismic intensity" after "at what level"

**Response:** Based on your suggestion, we have added "of seismic intensity" to add greater clarity.

10. Line 183: add "(Fig. 2)" after "Groups A and B"

**Response:** Done.

11. Lines 197-198: the sentence: "The most important component of a successful EEWS is a group of users who want alerts and can define the necessary capabilities of the system" is not clear

**Response:** We have revised this sentence to "The most important component of a successful EEWS is a group of users with awareness and preparedness, who want alerts and will take protective actions".

Would you please also provide an 'author's reply' to my comments and include a track changes document between the old manuscript and the new one (you can include this as part of your 'author's reply').

**Response:** As per your request, we have prepared a new version of the manuscript with the track-changes.

---

## Author Response (AR3)

October 5, 2021

Daniela Molinari, PhD
*Natural Hazards and Earth System Sciences*

Dear Dr. Molinari,

Thank you for accepting for publication our manuscript (nhess-2021-42), entitled "Effective earthquake early warning systems: Appropriate messaging and public awareness roles".

Based on your request, we have added the data availability section according to the manuscript preparation guidelines. Our manuscript is complete and includes the title, authors and their affiliations, abstract, tables, and figure and table captions. To promote our work, we have also provided a 500-character short summary.

We would like to thank you for the opportunity to publish our manuscript in *Natural Hazards and Earth System Sciences* as a "brief communication".

Please feel free to contact me with any questions or concerns you may have.

Sincerely,
* * *
Ya Tang, PhD

Department of Environment
College of Architecture and Environment
Sichuan University, Chengdu, China
tangya@scu.edu.cn